# Sexual dimorphism in trait variability and its eco-evolutionary and statistical implications

Susanne RK Zajitschek[1,2]*, Felix Zajitschek[1], Russell Bonduriansky[1], Robert C Brooks[1], Will Cornwell[1], Daniel S Falster[1], Malgorzata Lagisz[1], Jeremy Mason[3], Alistair M Senior[4], Daniel WA Noble[1,5†], Shinichi Nakagawa[1†]*

[1]Evolution & Ecology Research Center, School of Biological, Earth, and Environmental Sciences, University of New South Wales, Sydney, Australia; [2]Liverpool John Moores University, School of Biological and Environmental Sciences, Liverpool, United Kingdom; [3]European Bioinformatics Institute (EMBL-EBI), European Molecular Biology Laboratory, Wellcome Trust Genome Campus, Hinxton, United Kingdom; [4]University of Sydney, Charles Perkins Centre, School of Life and Environmental Sciences, School of Mathematics and Statistics, Sydney, Australia; [5]Division of Ecology and Evolution, Research School of Biology, Australian National University, Canberra, Australia

*For correspondence:
susi.zajitschek@gmail.com (SRKZ);
s.nakagawa@unsw.edu.au (SN)

†These authors contributed equally to this work

Competing interests: The authors declare that no competing interests exist.

**Abstract** Biomedical and clinical sciences are experiencing a renewed interest in the fact that males and females differ in many anatomic, physiological, and behavioural traits. Sex differences in trait variability, however, are yet to receive similar recognition. In medical science, mammalian females are assumed to have higher trait variability due to estrous cycles (the 'estrus-mediated variability hypothesis'); historically in biomedical research, females have been excluded for this reason. Contrastingly, evolutionary theory and associated data support the 'greater male variability hypothesis'. Here, we test these competing hypotheses in 218 traits measured in >26,900 mice, using meta-analysis methods. Neither hypothesis could universally explain patterns in trait variability. Sex bias in variability was trait-dependent. While greater male variability was found in morphological traits, females were much more variable in immunological traits. Sex-specific variability has eco-evolutionary ramifications, including sex-dependent responses to climate change, as well as statistical implications including power analysis considering sex difference in variance.

## Introduction

Sex differences arise because selection acts on the two sexes differently, especially on traits associated with mating and reproduction (*Darwin, 1871*). Therefore, sex differences are widespread, a fact which is unsurprising to any evolutionary biologist. However, scientists in many (bio-)medical fields have not necessarily regarded sex as a biological factor of intrinsic interest (*Clayton, 2016*; *Flanagan, 2014*; *Karp et al., 2017*; *Klein et al., 2015*; *Prendergast et al., 2014*; *Shansky and Woolley, 2016*). Therefore, many (bio-)medical studies have only been conducted with male subjects. Consequently, our knowledge is biased. For example, we know far more about drug efficacy in male compared to female subjects, contributing to a poor understanding of how the sexes respond differently to medical interventions (*Nowogrodzki, 2017*). This gap in knowledge is predicted to lead to overmedication and adverse drug reactions in women (*Zucker and Prendergast, 2020*). Only recently have (bio-)medical scientists started considering sex differences in their research (*Dorris et al., 2015*; *Ingvorsen et al., 2017*; *Robinson et al., 2017*; *Smarr et al., 2017*;

**eLife digest** Males and females differ in appearance, physiology and behavior. But we do not fully understand the health and evolutionary consequences of these differences. One reason for this is that, until recently, females were often excluded from medical studies. This made it difficult to know if a treatment would perform as well in females as males. To correct this, organizations that fund research now require scientists to include both sexes in studies. This has led to some questions about how to account for sex differences in studies.

One reason females have historically been excluded from medical studies is that some scientists assumed that they would have more variable responses to a particular treatment based on their estrous cycles. Other scientists, however, believe that males of a given species might be more variable because of the evolutionary pressures they face in competing for mates. Better understanding how males and females vary would help scientists better design studies to ensure they provide accurate answers.

Now, Zajitschek et al. debunk both the idea that males are more variable and the idea that females are more variable. To do this, Zajitschek et al. analyzed differences in 218 traits, like body size or certain behaviors, among nearly 27,000 male and female mice. This showed that neither male mice nor female mice were universally more different from other mice of their sex across all features. Instead, sex differences in how much variation existed in male or female mice depended on the individual trait. For example, males varied more in physical features like size, while females showed more differences in their immune systems.

The results suggest it is particularly important to consider sex-specific variability in both medical and other types of studies. To help other researchers better design experiments to factor in such variability, Zajitschek et al. created an interactive tool that will allow scientists to look at sex-based differences in individual features among male or female mice.

*Ahmad et al., 2017*; *Foltin and Evans, 2018*; *Thompson et al., 2018*). Indeed, the National Institutes of Health (NIH) have now implemented new guidelines for animal and human research study designs, requiring that sex be included as a biological variable (*Clayton, 2016*; *Clayton and Collins, 2014*; *NIH, 2015a*).

When comparing the sexes, biologists generally focus on mean differences in trait values, placing little or no emphasis on sex differences in trait variability (see *Figure 1* for a diagram explaining differences in means and variances). Despite this, two hypotheses exist that explain why trait variability might be expected to differ between the sexes. Interestingly, these two hypotheses make opposing predictions.

First, the 'estrus-mediated variability hypothesis' (*Figure 2*), which emerged in the (bio-)medical research field, assumes that the female estrous cycle (see e.g. *Prendergast et al., 2014*; *Beery and Zucker, 2011*) causes higher variability across traits in female subjects. A wide range of labile traits are presumed to co-vary with physiological changes that are induced by reproductive hormones. High variability is, therefore, expected to be particularly prominent when the stage of the estrous cycle is unknown and unaccounted for. This higher trait variability, resulting from females being at different stages of their estrous cycle, is the main reason for why female research subjects are often excluded from biomedical research trials, especially in the fields of neuroscience, physiology and pharmacology (*NIH, 2015a*). Female exclusion has traditionally been justified based on the grounds that including females in empirical research leads to a loss of statistical power, or that animals must be sampled across the estrous cycle for one to make valid conclusions, requiring more time and resources.

Second, the 'greater male variability hypothesis' suggests males exhibit higher trait variability because of two different mechanisms. The first mechanism is based on males being the heterogametic sex in mammals. Mammalian females possess two X chromosomes, leading to an 'averaging' of trait expression across the genes on each chromosome. In contrast, males exhibit greater variance because expression of genes on a single X chromosome is likely to lead to more extreme trait values (*Reinhold and Engqvist, 2013*). The second mechanism is based on males being under stronger sexual selection (*Pomiankowski and Moller, 1995*; *Cuervo and Møller, 1999*; *Cuervo and Møller,*

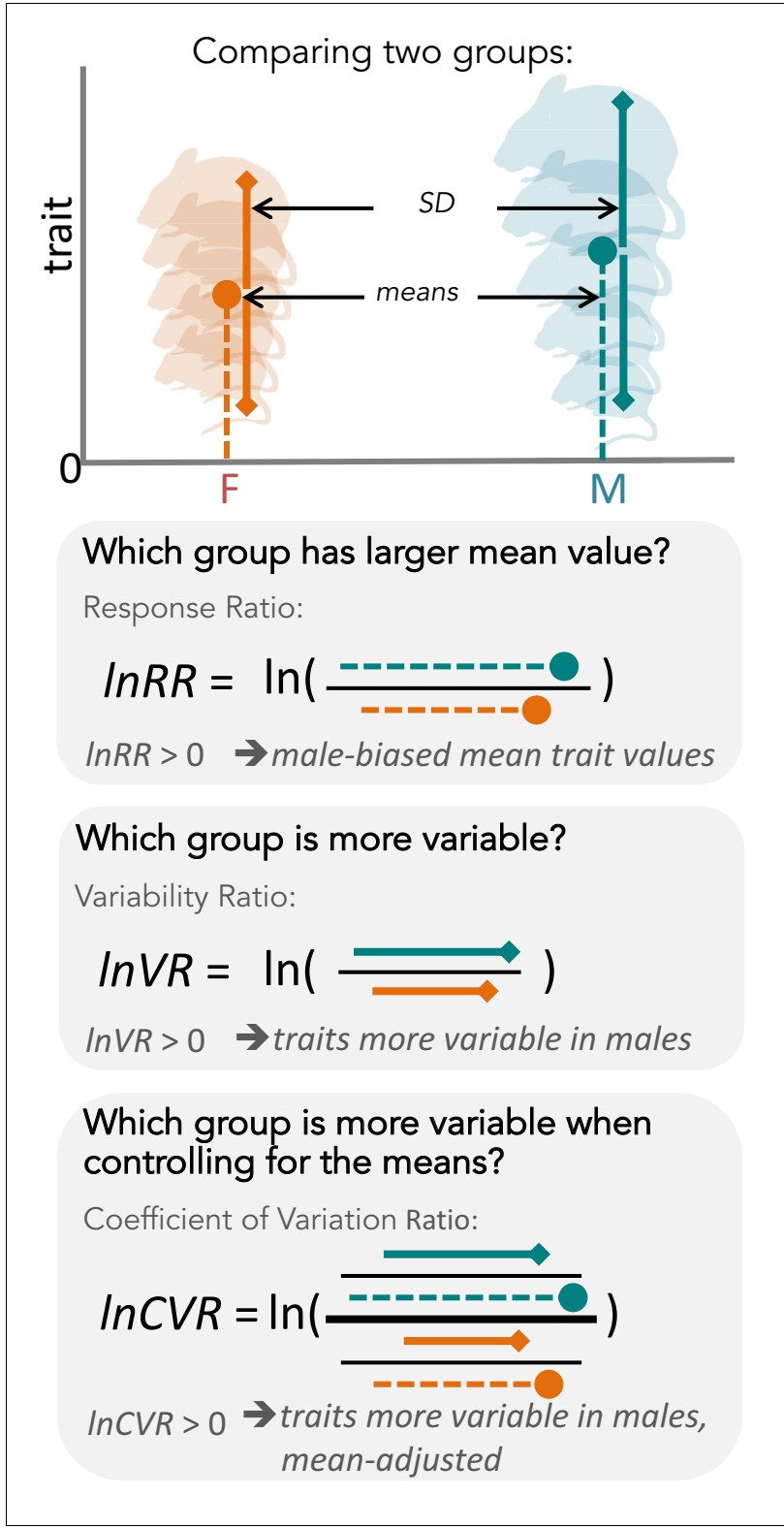

**Figure 1.** Overview of meta-analytic methods used to detect differences in means and variances in any given trait (e.g. body size in mice). The orange shading represents females (**F**), turquoise shading stands for males (**M**). The solid circle represents a mean trait value within the respective group. Solid lines represent standard deviation, with upper and lower bounds indicated by diamond shapes. Below, we present three types of effect sizes that can be used for comparing two groups, along with the respective formulas and interpretations. Compared to lnVR (the

*Figure 1 continued on next page*

*Figure 1 continued*

ratio of SD), lnCVR (the ratio of CV or relative variance) provides a more general measure of the difference in variability between two groups (mean-adjusted variability ratio).

The online version of this article includes the following figure supplement(s) for figure 1:

**Figure supplement 1.** Mean-variance relationships (log(Mean) vs log(SD, standard deviation)) across all traits for males (**A**) and females (**B**).

---

*2001*). Empirical evidence supports higher variability of traits that are sexually selected, often harbouring high genetic variance and being condition-dependent, which makes sense as 'condition' as a trait is likely to be based on numerous loci (*Rowe and Houle, 1996*; *Tomkins et al., 2004*). Thus, higher genetic and, thus, phenotypic variance resulting from sexual selection is expected to characterise sexually selected traits. In mammals, it is likely that both mechanisms are operating concomitantly. So far, the 'greater male variability hypothesis' has gained some support in the evolutionary and psychological literature (*Reinhold and Engqvist, 2013*; *Lehre et al., 2009*).

Here, we conduct the first comprehensive test of the greater male variability and estrus-mediated variability hypotheses in mice (*Figure 2*; *Reinhold and Engqvist, 2013*; *Johnson et al., 2008*;

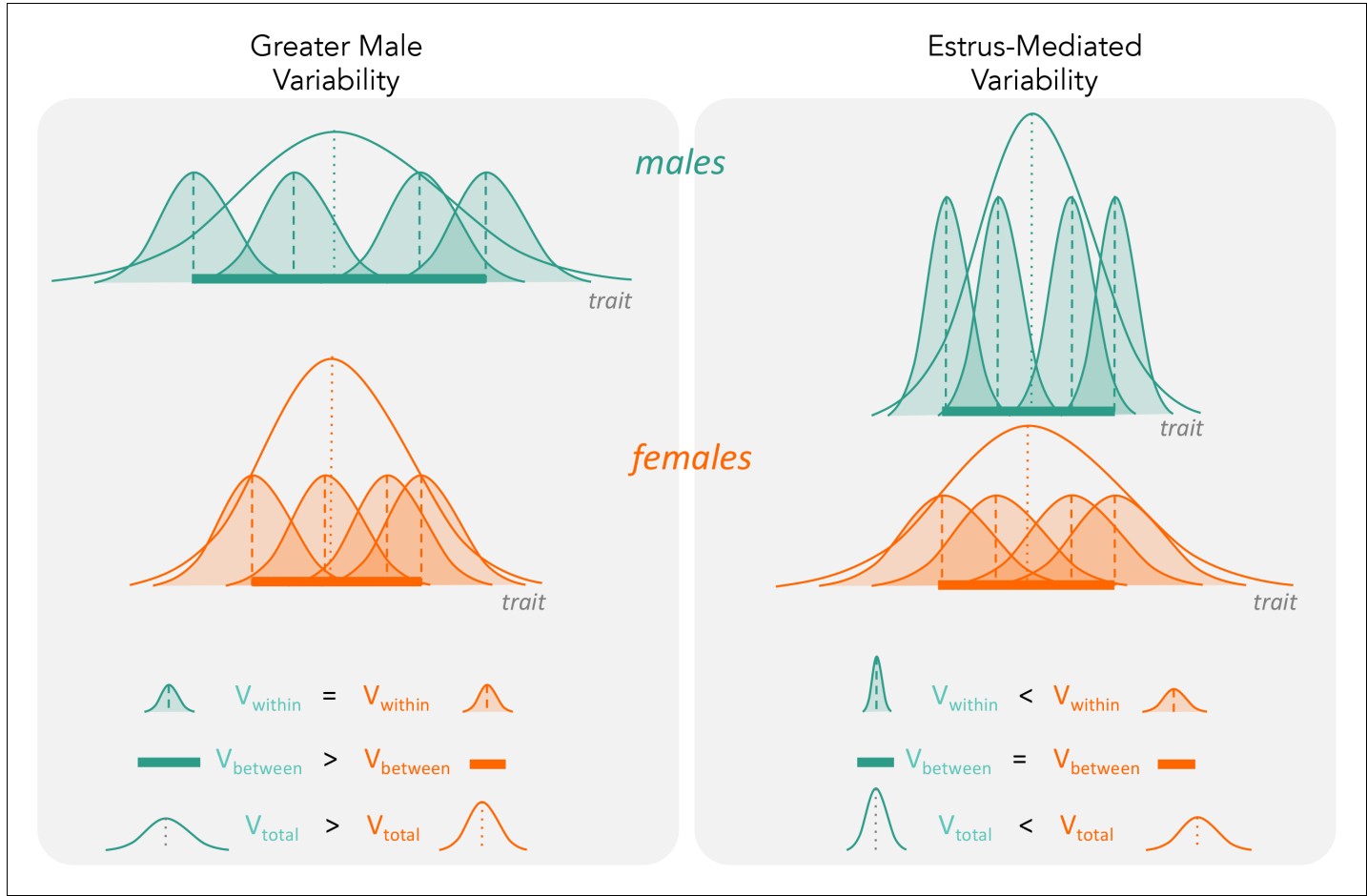

**Figure 2.** The two hypotheses ('greater male variability' versus 'estrus-mediated variability') have different predictions on how variabilities influence total observed phenotypic variance ($V_{total}$ in the figure). For greater male variability, the within-subject (or within-trait) variation $V_{within}$ could be potentially negligible or is equal in males and females. This is illustrated as the shaded distributions around each individual mean (dashed vertical lines), which are of equal area for the males (turquoise) and females (orange). The greater value of $V_{total}$ is driven by wider distribution of mean trait values in males compared to females (i.e. $V_{between}$, represented by a thick horizontal bar). The estrus-mediated variability hypothesis, in contrast, assumes that within-subject [or within-trait] variability is much higher in females than in males (broader orange-shaded trait distributions than turquoise distributions), while the variability of the means between individuals stays the same (thick horizontal bars).

*Hedges and Nowell, 1995*; *Itoh and Arnold, 2015*; *Becker et al., 2016*; *Beery, 2018*), examining sex differences in variance across 218 traits in 26,916 animals. To this end, we carry out a series of meta-analyses in two steps (*Figure 3*). First, we quantify the natural logarithm of the male to female coefficients of variation, CV, or relative variance (lnCVR) for each cohort (population) of mice, for different traits, along with the variability ratio of male to female standard deviations, SD, on the log scale (lnVR, following *Nakagawa et al., 2015*, see *Figure 1*). Then, we analyse these effect sizes to quantify sex bias in variance for each trait using meta-analytic methods. To better understand our results, and match them to previously reported sex differences in trait means (*Karp et al., 2017*), we also quantify and analyse the log response ratio (lnRR). Next, we statistically amalgamate the trait-level results to test our hypotheses and to quantify the degree of sex bias in and across nine

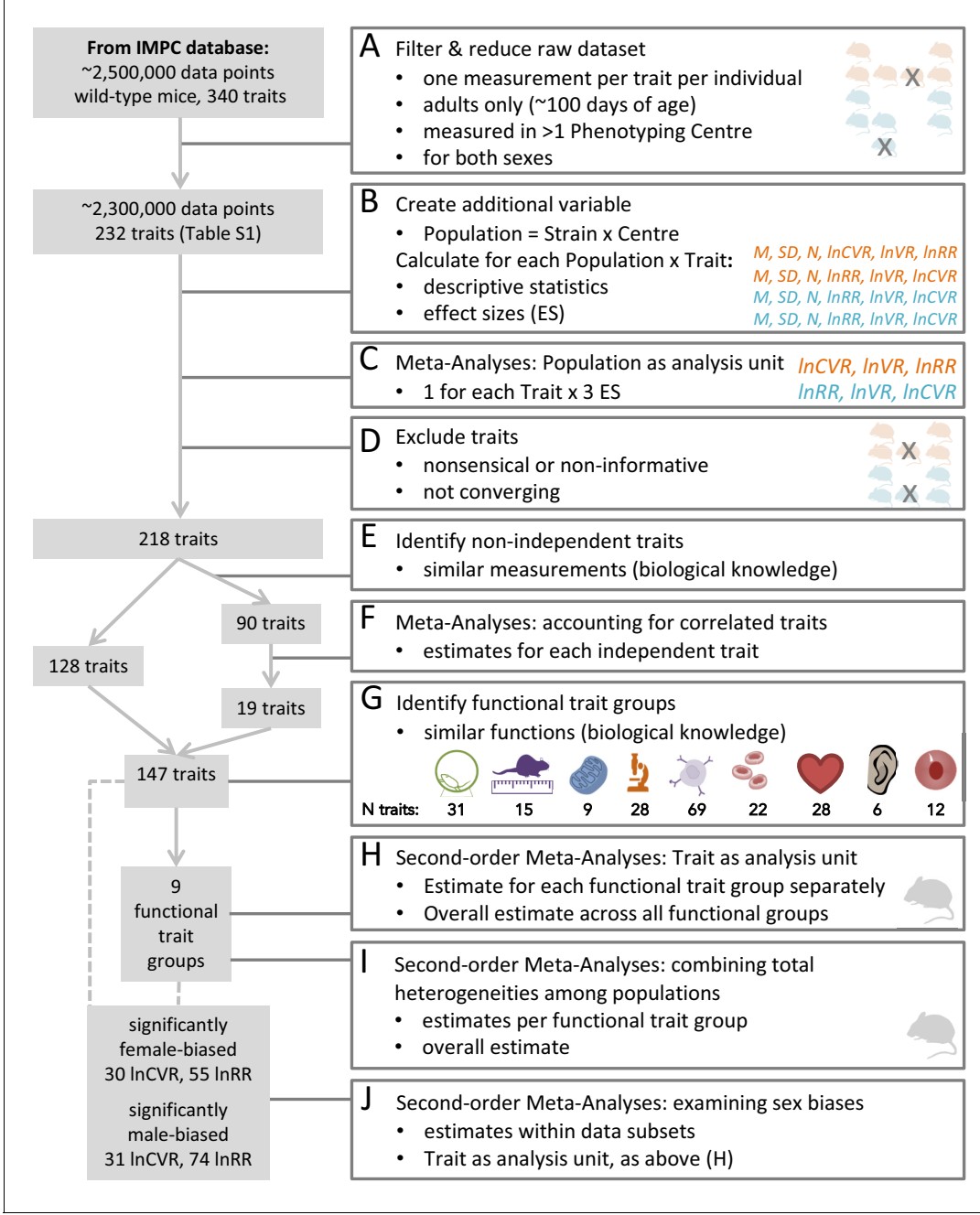

**Figure 3.** Workflow of data processing and meta-analysis.

functional trait groups (for details on the grouping, see below). Our meta-analytic approach allows easy interpretation and comparison with earlier and future studies. Further, the proposed method using lnCVR (and lnVR) is probably the only practical method to compare variability between two sexes within and across studies (*Nakagawa et al., 2015*; *Senior et al., 2020*), as far as we are aware. Also, the use of a ratio (i.e. lnRR, lnVR, lnCVR) between two groups (males and females) naturally controls for different units (e.g. cm, g, ml) as well as for changes in traits over time and space.

## Results

### Data characteristics and workflow

We used a dataset compiled by the International Mouse Phenotyping Consortium (*Dickinson et al., 2016*) (IMPC, dataset acquired 6/2018). To gain insight into systematic sex differences, we only included data of wildtype-strain adult mice, between 100 and 500 days of age. We removed cases with missing data, and selected measurements that were closest to 100 days of age (young adult) when multiple measurements of the same trait were available. To obtain robust estimates of sex differences, we only used data on traits that were measured in at least two different institutions (see workflow diagram, *Figure 3*).

Our dataset comprised 218 continuous traits (after initial data cleaning and pre-processing; *Figure 3*). It contains information from 26,916 mice from nine wildtype strains that were studied across 11 institutions. We combined mouse strain/institution information to create a biological grouping variable (referred to as 'population' in *Figure 3B*; see also *Supplementary file 1*, Table 1 for details), and the mean and variance of a trait for each population was quantified. We assigned traits according to related procedures into functionally and/or procedurally related trait groups to enhance interpretability (referred to as 'functional groups' hereafter; see also *Figure 3G*). Our nine functional trait groups were: behaviour, morphology, metabolism, physiology, immunology, hematology, heart, hearing and eye (for the rationale of these functional groups and related details, see Methods and *Supplementary file 1*, Table 3).

### Testing the two hypotheses

We found that some means and variabilities of traits were biased towards males (i.e. 'male-biased', hereafter; turquoise shaded traits, *Figure 4*), but others towards females (i.e. 'female-biased', hereafter; orange shading, *Figure 4*) within all functional groups. These sex-specific biases occur in mean trait sizes and also in our measures of trait variability. There were strong positive relationships between mean and variance across traits (r > 0.94 on the log scale; *Figure 1—figure supplement 1*), and therefore, we report the results of lnCVR, which controls for differences in means, in the main text. Results on lnVR are presented as figure supplements (*Figure 4—figure supplements 1* and *2*).

There was no consistent pattern in which sex has more variability (lnCVR) in the examined traits (left panel in *Figure 4A*). Our meta-analytic results also did not support a consistent pattern of either higher male variability or higher female variability (see *Figure 4B*, left panel: 'All' indicates that across all traits and functional groups, there was no significant sex bias in variances; lnCVR = 0.005, 95% confidence interval, 95% CI = [−0.009 to 0.018]). However, there was high heterogeneity among traits ($I^2$ = 76.5%, *Supplementary file 1*, Table 4 and see also Table 5), indicating sex differences in variability are trait-dependent, corroborating our general observation that variability in some traits was male-biased but others female-biased (*Figure 4A*).

As expected, specific functional trait groups showed significant sex-specific bias in variability (*Figure 4B*). The variability among traits within a functional group was lower than that of all the traits combined (*Supplementary file 1*, Table 4). For example, males exhibited an 8.05% increase in CV relative to females for morphological traits (lnCVR = 0.077; CI = [0.041 to 0.113], $I^2$ = 67.3%), but CV was female-biased for immunological traits (6.59% higher in females, lnCVR = −0.068, CI = [−0.098 to 0.038], $I^2$ = 40.8%) and eye morphology (7.85% higher in females, lnCVR = −0.081, CI = [−0.147 to (−0.016)], $I^2$ = 49.8%).

The pattern was similar for overall sexual dimorphism in mean trait values (here, a slight male bias is indicated by larger 'turquoise' than 'orange' areas; *Figure 4B*, right and *Figure 4B*, lnRR: 'All', lnRR = 0.012, CI = [−0.006 to 0.31]). Trait means (lnRR) were 7% larger for males (lnRR = 0.067; CI =

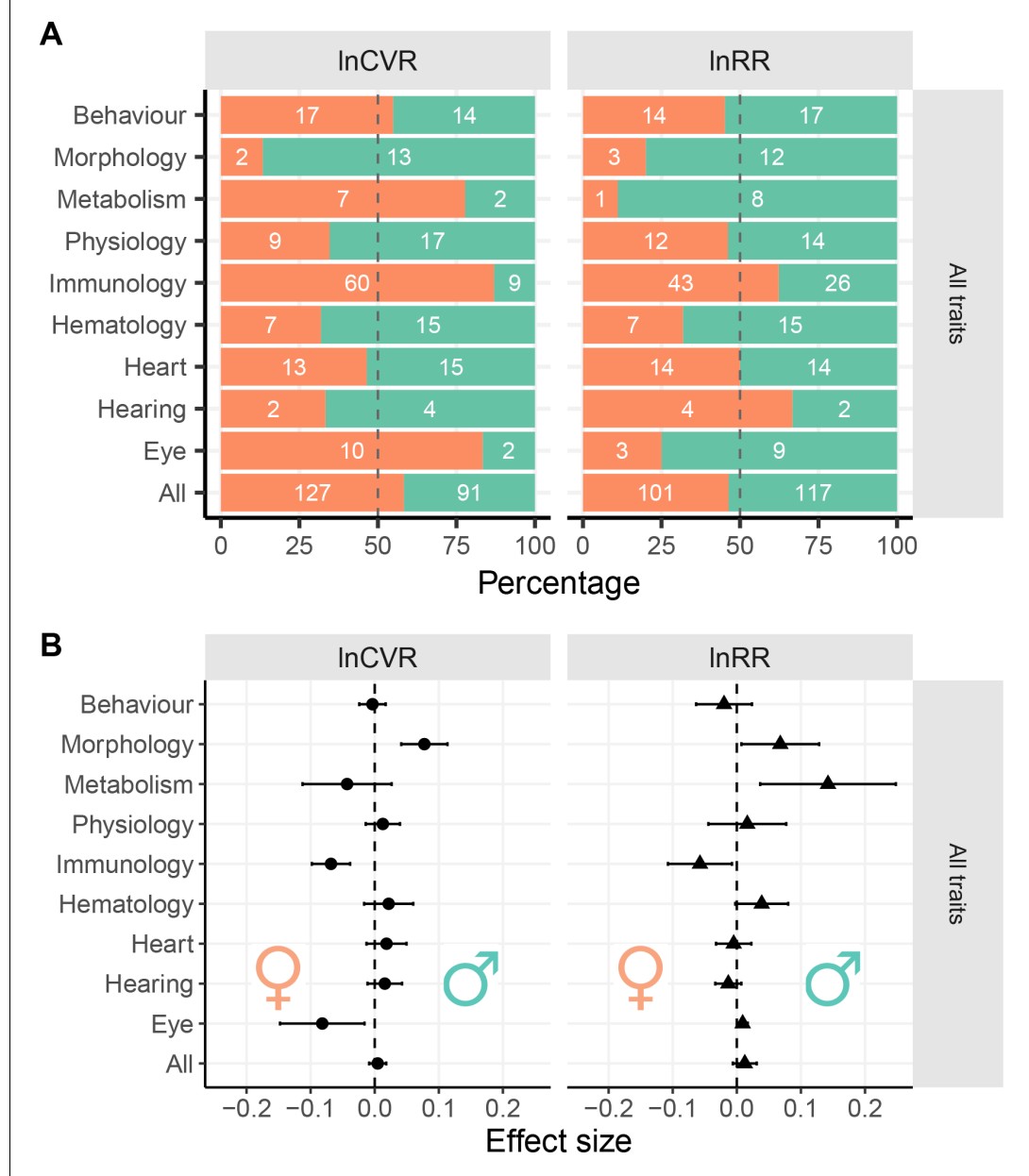

**Figure 4.** Sex bias in trait groups. Panel (**A**) shows the numbers of traits that are either male-biased (turquoise) or female-biased (orange) across functional groups. The x-axes in Panel A represents the overall percentages of traits with a given direction of sex bias: orange shading when meta-analytic mean < 0 (female-biased), turquoise shading when meta-analytic mean > 0 (male-biased). White numbers inside the turquoise bars represent numbers of traits that show male bias within a given group of traits, numbers inside the orange bars represent the number of female-biased traits. Panel (**B**) shows effect sizes and 95% CI from separate meta-analysis for each functional group (**Figure 3H**). Traits that are male-biased in Panel B are shifted towards the righthand side of the zero-midline (near the turquoise male symbol), whereas female-biased traits are shifted towards the left (near the orange female symbol).

The online version of this article includes the following source data and figure supplement(s) for figure 4:

**Source data 1.** Numbers of sex-biased traits.

**Source data 2.** Effect sizes of sex bias in functional groups.

**Figure supplement 1.** Percentages and numbers of either male (turqoise bars) or female (orange bars) biased traits (Panel A) across functional groups, this time for lnCVR (left hand side), lnVR (middle) and lnRR (right hand side).

**Figure supplement 2.** Sex bias in trait groups for lnCVR, lnVR and lnRR.

[0.007 to 0.128]) in morphological traits and 15.3% larger in males for metabolic traits (lnRR = 0.142; CI = [0.036 to 0.248]). In contrast, females had 5.59% (lnRR = 0.057, CI = [−0.107 to (−0.007)]) larger means than those of males for immunological traits. We note that these meta-analytic estimates were accompanied by very large between-trait heterogeneity values (morphology $I^2$ = 99.7%, metabolism $I^2$ = 99.4%, immunology $I^2$ = 96.2; see *Supplementary file 1*, Table 4), indicating that even within the same functional groups, the degree and direction of sex bias in the mean was not consistent among traits.

## Discussion

We tested competing predictions from two hypotheses explaining why sex biases in trait variability exist. Neither the 'greater male variability' hypothesis nor the 'estrus-mediated variability' hypothesis explain the observed patterns in sex-biased trait variation on their own. Therefore, our results add further empirical weight to calls that question the basis for the routine exclusion of one sex in bio-medical research based on the estrus-mediated variability hypothesis (*Flanagan, 2014*; *Klein et al., 2015*; *Prendergast et al., 2014*; *Shansky and Woolley, 2016*; *Becker et al., 2016*). It is important to know that for each trait we estimated the mean effect size (i.e. lnCVR) over strains and locations. As such, our results may not necessarily apply to every group of mice, which may or may not result in stronger support for either of the two hypotheses.

### Greater male variability vs. estrus-mediated variability?

Evolutionary biologists commonly expect greater variability in the heterogametic sex than the homo-gametic sex. In mammals, males are heterogametic, and hence are expected to exhibit higher trait variability compared to females, which is also consistent with an expectation from sexual selection theory (*Reinhold and Engqvist, 2013*). Our results provide only partial support for the greater male variability hypothesis, because the expected pattern only manifested for morphological traits (see *Figures 4* and *5*). This result corroborates a previous analysis across animals, which found that the heterogametic sex was more variable in body size (*Reinhold and Engqvist, 2013*). However, our data do not support the conclusion that higher variability in males occurs across all traits, including for many other morphological traits.

The estrus-mediated variability hypothesis was, at least until recently (*Prendergast et al., 2014*; *Smarr et al., 2017*), regularly used as a rationale for including only male subjects in many biomedical studies. So far, we know very little about the relationship between hormonal fluctuations and general trait variability within and among female subjects. Our results are consistent with the estrus-mediated variability hypothesis for immunological traits only. Immune responses can strongly depend on sex hormones (*Zuk and McKean, 1996*; *Grossman, 1989*), which may explain higher female variability in these traits. However, if estrus status affects traits through variation in hormone levels, we would expect to also find higher female variability in physiological and hematological traits. This was not the case in our dataset. Interestingly, however, eye morphology (structural traits, which should fluctuate little across the estrous cycle) also appeared to be more variable in females than males, but little is known about sex differences in ocular traits in general (*Wagner et al., 2008*; *Shaqiri et al., 2018*). Overall, we find no consistent support for the female estrus-mediated variability hypothesis.

In line with our findings, recent studies have refuted the prediction of higher female variability (*Prendergast et al., 2014*; *Smarr et al., 2017*; *Beery and Zucker, 2011*; *Becker et al., 2016*; *Beery, 2018*). For example, several rodent studies have found that males are more variable than females (*Prendergast et al., 2014*; *Smarr et al., 2017*; *Becker et al., 2016*; *Beery, 2018*; *Fritz et al., 2017*; *Mogil and Chanda, 2005*). Further studies should investigate whether higher female variability in immunological traits is indeed due to the estrous cycle, or generally because of greater between-individual variation (*Figure 2*).

In general, we found many traits to be sexually dimorphic (*Figure 5*) in accordance with the previous study, which used the same database (*Karp et al., 2017*). Although the original study also provided estimates for sex differences in traits both with and without controlling for weight (we did not control for weight; *Nakagawa et al., 2017*). More specifically, males are larger than females, while females have higher immunological parameters (see *Figure 5*). Notably, the most sexually dimorphic trait means also show the greatest differences in trait variance (*Figures 4* and *5*). Indeed, theory predicts that sexually selected traits (e.g. larger body size for males due to male-male competition) are

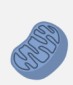

## Behaviour
➔ *few sex-biased mean trait values*
➔ *little sex-bias in trait variability*

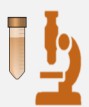

## Morphology
➔ **mostly male-biased mean trait values**
➔ **traits often more variable in males**

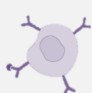

## Metabolism
➔ *mostly male-biased mean trait values*
➔ *little sex-bias in trait variability*

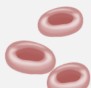

## Physiology
➔ *few sex-biased mean trait values*
➔ *little sex-bias in trait variability*

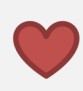

## Immunology
➔ **mostly female-biased mean trait values**
➔ **traits often more variable in females**

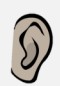

## Hematology
➔ *few sex-biased mean trait values*
➔ *little sex-bias in trait variability*

## Heart
➔ *few sex-biased mean trait values*
➔ *little sex-bias in trait variability*

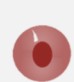

## Hearing
➔ *few sex-biased mean trait values*
➔ *little sex-bias in trait variability*

## Eye
➔ *few sex-biased mean trait values*
➔ **traits often more variable in females**

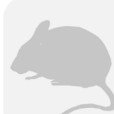

## All traits
➔ *few sex-biased mean trait values*
➔ *little sex-bias in trait variability*

**Figure 5.** Summary of sex differences in the mean trait values (lnRR) and variances (lnCVR) across nine functional trait groups, and overall.

likely more variable, as these traits are often condition-dependent (*Rowe and Houle, 1996*). Therefore, this sex difference in variability could be more pronounced under natural conditions compared to laboratory settings. This relationship may explain why male-biased morphological traits are larger and more variable.

## Eco-evolutionary implications

We have used lnCVR values to compare phenotypic variability (CV) between the sexes. When lnCVR is used for fitness-related traits, it can signify sex differences in the 'opportunity for selection' between females and males (*Rowe and Houle, 1996*). If we assume that phenotypic variation (i.e. variability in traits) has a heritable basis, then large ratios of lnCVR may indicate differences in the evolutionary potential of each sex to respond to selection, at least in the short term (*Hansen and Houle, 2008*). For example, more variable morphological traits of males could potentially provide them with better capacity than females to adapt morphologically to a changing climate. We note, however, that in our study, lnCVR reflects sex differences in trait variability within strains, such that the variability differences we observe between the sexes may be partially the result of phenotypic plasticity.

Demographic parameters, such as age-dependent mortality rate (*Lemaître et al., 2020*) can often be different for each sex. For example, a study on European sparrowhawks found that variability in mortality was higher in females compared to males (*Colchero et al., 2017*). In this species, sex-specific variation affects age-dependent mortality and results in higher average female life expectancy. Therefore, population dynamic models, which make predictions about how populations change in their size over time, should take sex differences in variability into account to produce more accurate predictions (*Caswell and Weeks, 1986*; *Lindström and Kokko, 1998*). In our rapidly changing world, better predictions on population dynamics are vital for understanding whether climate change is likely to result in population extinction and lead to further biodiversity loss.

## Statistical and practical implications

It is now mandatory to include both sexes in biomedical experiments and clinical trials funded by the NIH, unless there exists strong justification against the inclusion of both sexes (*NIH, 2015a*; *NIH, 2015b*). In order to conduct meaningful research and make sound clinical recommendations for both male and female patients, it is necessary to understand both how trait means and variances differ between the sexes. If one sex is systematically more variable in a trait of interest than the other, then experiments should be designed to accommodate relative differences in statistical power between the sexes (which has not been considered before, see *Flanagan, 2014*; *Klein et al., 2015*; *Prendergast et al., 2014*; *Shansky and Woolley, 2016*). For example, female immunological traits are generally more variable (i.e. having higher CV and SD). Therefore, in an experiment measuring immunological traits, we would need to include a larger sample ($N$) of females than males ($N_{[female]} > N_{[male]}$; $N_{[total]} = N_{[female]} + N_{[male]}$) to achieve the same power as when the experiment only includes males ($N_{[total*]} = 2N_{[male]}$). In other words, in an experiment with both sexes we would need a larger sample size than the same experiment with males only ($N_{[total]} > N_{[total*]}$).

To help researchers adjust their sex-specific sample size to achieve optimal statistical power, we provide an online tool (ShinyApp; https://bit.ly/sex-difference). This tool may serve as a starting point for checking baseline variability for each sex in mice. The sex bias (indicated by the % difference between the sexes) is provided for separate traits, procedures, and functional groups. These meta-analytic results are based on our analyses of more than 2 million rodent data points, from 26,916 individual mice. We note, however, that variability in a trait measured in untreated individuals maintained under carefully standardized environmental conditions, as reported here, may not directly translate into the same variability when measured in experimentally treated individuals, or individuals exposed to a range of environments (i.e. natural populations or human cohorts). Further, these estimates are overall mean differences across strains and locations. Therefore, these may not be particularly informative if one's experiment only includes one specific strain. Nonetheless, we point out that our estimates may be useful in the light of a recent recommendation of using 'heterogenization' where many different strains are systematically included (i.e. randomized complete block design) to increase the robustness of experimental results (*Voelkl et al., 2020*). However, note that an experiment with heterogenization might only include a few strains with several animals per strain.

Even in such a case, using just a few strains, our tool could provide potentially useful benchmarks. Incidentally, heterogenization would be key to making one's experimental outcome more generalizable (*Webster and Rutz, 2020*).

Importantly, when two groups (e.g. males and females) show differences in variability, we violate homogeneity of variance or homoscedasticity assumptions. Such a violation is detrimental because it leads to a higher Type I error rate. Therefore, we should consider incorporating heteroscedasticity (different variances) explicitly or using robust estimators of variance (also known as 'the sandwich variance estimator') to prevent an inflated Type I error rate (*Cleasby and Nakagawa, 2011*), especially when we compare traits between the sexes.

## Conclusion

We have shown that sex biases in variability occur in many mouse traits, but that the directions of those biases differ between traits. Neither the 'greater male variability' nor the 'estrus-mediated variability' hypothesis provides a general explanation for sex differences in trait variability. Instead, we have found that the direction of the sex bias varies across traits and among trait types (*Figures 4* and *5*). Our findings have important ecological and evolutionary ramifications. If the differences in variability correspond to the potential of each sex to respond to changes in specific environments, this sex difference needs to be incorporated into demographic and population dynamic modelling. Moreover, in the (bio-)medical field, our results should inform decisions during study design by providing more rigorous power analyses that allow researchers to incorporate sex-specific differences for sample size. We believe that taking sex differences in trait variability into account will help avoid misleading conclusions and provide new insights into sex differences across many areas of biological and bio-medical research. Ultimately, such considerations will not only better our knowledge, but also close the current gaps in our biased knowledge (*Tannenbaum et al., 2019*).

## Materials and methods

### Data selection and process

The IMPC (International Mouse Phenotyping Consortium) provides a comprehensive catalogue of mammalian gene function for investigating the genetics of health and disease, by systematically collecting phenotypes of knock-out and wildtype mice. To investigate differences in trait variability between the sexes, we only considered the data for wildtype control mice. We retrieved the dataset from the IMPC server in June 2018 and filtered it to contain non-categorical traits for wildtype mice. The initial dataset comprised over 2,500,000 data points for 340 traits. In cases where multiple measurements were taken over time, data cleaning started with selecting single measurements for each individual and trait. In these cases, we selected the measurement closest to '100 days of age'. All data are from unstaged females (with no information about the stage of their estrous cycle). We excluded data for juvenile and unsexed mice (*Figure 3A*; this dataset and scripts can be found on https://rpubs.com/SusZaj/ESF; https://bit.ly/code-mice-sex-diff; raw data: https://doi.org/10.5281/zenodo.3759701).

### Grouping and effect-size calculation

We created a grouping variable called 'population' (*Figure 3B*). A population comprised a group of individuals belonging to a distinct wildtype strain maintained at one particular location (institution); populations were identified for every trait of interest. Our data were derived from 11 different locations/institutions, and a given location/institution could provide data on multiple populations (see *Supplementary file 1*, Table 1 for details on numbers of strains and institutions). We included only populations that contained data points for at least six individuals, and which had information for members of both sexes; further, populations for a particular trait had to come from at least two institutions to be eligible for inclusion. After this selection process, the dataset contained 2,300,000 data points across 232 traits. Overall, we meta-analysed traits with between 2–18 effect sizes (mean = 9.09 effects, SD = 4.47). However, each meta-analysis contained a total number of individual mice that ranged from 83/91 to 13467/13449 (males/females). While a minimum of N = 6 mice were used to create effect sizes for any given group (male or female), in reality samples sizes of male/female groups were much larger (males: mean = 396.66 (SD = 238.23), median = 465.56;

females: mean = 407.35 (SD = 240.31), median = 543.89). We used the function *escalc* in the R package, *metafor* (*Viechtbauer, 2010*) to obtain lnCVR, lnVR and lnRR and their corresponding sampling variance for each trait for each population; we worked in the R environment for data cleaning, processing and analyses (*R Development Core Team, 2017*, version 3.6.0; for the versions of all the software packages used for this article and all the details and code for the statistical analyses, see *Source code 1* and repositories). As mentioned above, the use of ratio-based effect sizes, such as lnCVR, lnVR and lnRR, controls for baseline changes over time and space, assuming that these changes affect males and females similarly. However, we acknowledge that we could not test this assumption.

## Meta-analyses: overview

We conducted meta-analyses at two different levels (*Figure 3C–J*). First, we conducted a meta-analysis for each trait for all three effect-size types (lnRR, lnVR and lnCVR), calculated at the 'population' level (i.e. using population as a unit of analysis). Second, we statistically amalgamated overall effect sizes estimated at each trait (i.e. overall trait means as a unit of analysis) after accounting for dependence among traits. In other words, we conducted second-order meta-analyses (*Nakagawa et al., 2019*). We used the second-order meta-analyses for three different purposes: (A) estimating overall sex biases in variance (lnCVR and lnVR) and mean (lnRR) in the nine functional groups (for details, see below) and in all these groups combined (the overall estimates); (B) visualizing heterogeneities across populations for the three types of effect size in the nine functional trait groups, which complemented the first set of analyses (*Figure 3I*, Table 6 in *Supplementary file 1*); and (C) when traits were found to be significantly sex-biased, grouping such traits into either male-biased and female-biased traits, and then, estimating overall magnitudes of sex bias for both sexes again for the nine functional trait groups. Only the first second-order meta-analysis (A) directly related to the testing of our hypotheses, results of B and C are found in *Supplementary file 1* and figures and reported in our freely accessible code.

## Meta-analyses: population as an analysis unit

To obtain degree of sex bias for each trait mean and variance (*Figure 3C*), we used the function *rma.mv* in the R package *metafor* (*Viechtbauer, 2010*) by fitting the following multilevel meta-analytic model, an extension of random-effects models (sensu *Nakagawa and Santos, 2012*):

$ES_i$ ~ 1 + (1 | $Strain_j$) + (1 | $Location_k$) + (1 | $Unit_i$) + $Error_i$, where '$ES_i$' is the $i$th effect size (i.e. lnCVR, lnVR and lnRR) for each of 232 traits, the '1' is the overall intercept (other '1's are random intercepts for the following random effects), '$Strain_j$' is a random effect for the $j$th strain of mice (among nine strains), '$Location_k$' is a random effect for the $k$th location (among 11 institutions), '$Unit_i$' is a residual (or effect-size level or 'population-level' random effect) for the $i$th effect size, '$Error_i$' is a random effect of the known sampling error for the $i$th effect size. Given the model above, meta-analytic results had two components: (1) overall means with standard errors (95% confidence intervals), and (2) total heterogeneity (the sum of the three variance components, which is estimated for the random effects). Note that overall means indicate average (marginalised) effect sizes over different strains and locations, and total heterogeneities reflect variation around overall means due to different strains and locations.

We excluded traits which did not carry useful information for this study (i.e. fixed traits, such as number of vertebrae, digits, ribs and other traits that were not variable across wildtype mice; note that this may be different for knock-down mutant strains) or where the meta-analytic model for the trait of interest did not converge, most likely due to small sample size from the dataset (14 traits, see SI Appendix, for details: Meta-analyses; 1. Population as analysis unit). We therefore obtained a dataset containing meta-analytic results for 218 traits, at this stage, to use for our second-order meta-analyses (*Figure 3D*).

## Meta-analyses: accounting for correlated traits

Our dataset of meta-analytic results included a large number of non-independent traits. To account for dependence, we identified 90 out of 218 traits, and organized them into 19 trait sub-groups (containing 2–10 correlated traits, see *Figure 3E*). For example, many measurements (i.e. traits) from hematological and immunological assays were hierarchically clustered or overlapped with each other

(e.g. cell type A, B and A+B). We combined the meta-analytic results from 90 traits into 19 meta-analytic results (*Figure 3F*) using the function *robu* in the R package *robumeta* with the assumption of sampling errors being correlated with the default value of $r$ = 0.8 (*Fisher et al., 2017*). Consequently, our final dataset for secondary meta-analyses contained 147 traits (i.e. the newly condensed 19 plus the remaining 128 independent traits, see *Figure 3*, *Supplementary file 1*, Table 2), which we assume to be independent of each other.

## Second-order meta-analyses: trait as an analysis unit

We created our nine overarching functional groups of traits (*Figure 3G*) by condensing the IMPC's 26 procedural categories ('procedures') into related clusters. The categories were based on procedures that were biologically related, in conjunction with measurement techniques and the number of available traits in each category (see *Supplementary file 1*, Table 3 for a list of clustered traits, procedures and grouping terms). To test our two hypotheses about how trait variability changes in relation to sex, we estimated overall effect sizes for nine functional groups by aggregating meta-analytic results via 'classical' random-effect models using the function rma.uni in the R package *metafor* (*Viechtbauer, 2010*). In other words, we conducted three sets of 10 second-order meta-analyses (i.e. meta-analyzing 3 types of effect size: lnRR, lnVR and lnCVR for nine functional groups and one for all the groups combined, *Figure 3H*). Although we present the frequencies of male- and female-biased traits in *Figure 4A*, we did not run inferential statistical tests on these counts because such tests would be considered as vote-counting, which has been severely criticised in the meta-analytic literature (*Higgins, 2019*).

## Acknowledgements

SRKZ and ML were supported by the Australian (ARC) Discovery Grant (DP180100818) awarded to SN. JM was supported by EMBL core funding and the NIH Common Fund (UM1-H G006370). AMS was supported by an ARC fellowship (DE180101520).

## Additional information

### Funding

| Funder | Grant reference number | Author |
| --- | --- | --- |
| Australian Research Council | DP180100818 | Shinichi Nakagawa |
| National Institutes of Health | UM1-H G006370 | Jeremy Mason |
| Australian Research Council | DE180101520 | Alistair M Senior |
| Australian Research Council | FT160100113 | Daniel S Falster |

The funders had no role in study design, data collection and interpretation, or the decision to submit the work for publication.

### Author contributions

Susanne RK Zajitschek, Formal analysis, Visualization, Writing - original draft, Writing - review and editing, Project administration; Felix Zajitschek, Formal analysis, Investigation, Visualization, Writing - review and editing; Russell Bonduriansky, Robert C Brooks, Investigation, Writing - review and editing; Will Cornwell, Investigation, Methodology, Writing - review and editing; Daniel S Falster, Alistair M Senior, Formal analysis, Methodology, Writing - review and editing; Malgorzata Lagisz, Visualization, Methodology, Writing - review and editing; Jeremy Mason, Resources, Data curation, Writing - review and editing; Daniel WA Noble, Validation, Methodology, Writing - review and editing; Shinichi Nakagawa, Conceptualization, Resources, Supervision, Funding acquisition, Methodology, Project administration, Writing - review and editing, Writing - original draft

## Author ORCIDs

Susanne RK Zajitschek (iD) https://orcid.org/0000-0003-4676-9950
Felix Zajitschek (iD) http://orcid.org/0000-0001-6010-6112
Russell Bonduriansky (iD) https://orcid.org/0000-0002-5786-6951
Robert C Brooks (iD) https://orcid.org/0000-0001-6926-0781
Will Cornwell (iD) https://orcid.org/0000-0003-4080-4073
Daniel S Falster (iD) https://orcid.org/0000-0002-9814-092X
Malgorzata Lagisz (iD) https://orcid.org/0000-0002-3993-6127
Jeremy Mason (iD) https://orcid.org/0000-0002-2796-5123
Alistair M Senior (iD) http://orcid.org/0000-0001-9805-7280
Daniel WA Noble (iD) https://orcid.org/0000-0001-9460-8743
Shinichi Nakagawa (iD) https://orcid.org/0000-0002-7765-5182

## Decision letter and Author response

Decision letter https://doi.org/10.7554/eLife.63170.sa1
Author response https://doi.org/10.7554/eLife.63170.sa2

---

# Additional files

## Supplementary files

• Source code 1. This markdown file contains all steps from processing the raw data file through to meta-analyses to Figure and table generation. The knitted html version can be viewed at https://rpubs.com/SusZaj/ESF.

• Supplementary file 1. Supplementary tables. Table 1. Summary of the available numbers of male and female mice from each strain and originating institution Table 2. Trait categories (parameter_group) and the number of correlated traits within these categories. Traits were meta-analysed using robumeta Table 3. We use this corrected (for correlated traits) results table, which contains each of the meta-analytic means for all effect sizes of interest, for further analyses. We further use this table as part of the Shiny App, which is able to provide the percentage differences between males and females for mean, variance and coefficient of variance (continued below) (Table 4). Summary of overall meta-analyses on the functional trait group level (GroupingTerm). Results for lnCVR, lnVR and lnRR and their respective upper and lower 95 percent CI's, standard error and $I^2$ values are provided. Values truncated at five decimal places for readability. Table 5. Provides an overview of meta-analysis results performed on traits that were significantly biased towards either sex. This table summarizes findings for both sexes and the respective functional trait groups. Values truncated at five decimal places for readability. Table 6. Summarizes our findings on heterogeneity due to institutions and mouse strains. These results are based on meta-analyses on sigma$^2$ and errors for mouse strains and centres (Institutions), following the identical workflow from above. Values truncated at five decimal places for readability.

• Transparent reporting form

## Data availability

The code and data generated during this study are freely accessible on GitHub. (https://github.com/itchyshin/mice_sex_diff; copy archived at https://archive.softwareheritage.org/swh:1:rev:2868f59b32d05a61091e70962e6e6a16463c6a64/) as well as OSF (https://osf.io/25h4t/). Original/source data (pre-cleaned dataset as downloaded from IMPC) can be downloaded from zenodo (DOI: 10.5281/zenodo.3759701). The supporting files also contain the full code workflow.

The following dataset was generated:

| Author(s) | Year | Dataset title | Dataset URL | Database and Identifier |
|---|---|---|---|---|
| Zajitschek SRK, Zajitschek F, Bonduriansky R, Brooks RC, Cornwell W, | 2020 | Raw data for: Sex and Power: Sexual dimorphism in trait variability and its eco-evolutionary and statistical implications | https://doi.org/10.5281/zenodo.3759701 | Zenodo, 10.5281/zenodo.3759701 |

Falster DS, Lagiz M, Mason J, Senior AM, Noble DWA, Nakagawa S

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
