## [Decision Letter]

**Acceptance summary:**

This study makes a valuable contribution to our understanding of sex-differences in trait variability. It applies a meta-analytic approach to mouse datasets to test two hypotheses of sex-specific variability in mammals: that females show greater variability due to estrous, and that males show greater variability due to heterogamy and/or sexual selection. It reveals that, at least for mice, neither hypothesis is universally true for all traits. Rather, variability is greater in females for some traits, and greater in males for others. These interesting results provide new insights into sex bias in variability, that will inform experimental design in diverse biological fields.

**Decision letter after peer review:**

[Editors’ note: the authors submitted for reconsideration following the decision after peer review. What follows is the decision letter after the first round of review.]

Thank you for submitting your work entitled "Sex and Power: sexual dimorphism in trait variability and its eco-evolutionary and statistical implications" for consideration by *eLife*. Your article has been reviewed by three peer reviewers, including Rosalyn Gloag as the Reviewing Editor and Reviewer #1, and the evaluation has been overseen by a Senior Editor. The following individual involved in the review of your submission has agreed to reveal their identity: Irving Zucker (Reviewer #3).

Our decision has been reached after consultation between the reviewers. Based on these discussions, and the individual reviews appended below, we regret to inform you that your work in its current form is not suitable for publication in *eLife*.

All reviewers agreed that the topic of the study was an interesting one, and that the issue of sex differences in trait variability is relevant to good experimental design. As you'll see below, however, reviewer #2 felt that the current analytical treatment of this mouse dataset is not appropriate to the question. Of particular concern is that sources of variability other than sex were not adequately considered. We recommend that this issue, and others outlined in the reviews below, are carefully addressed in any revision. Given the interest in the topic, we are prepared to reconsider a thoroughly revised version of the manuscript, if you think you can adequately address the concerns raised, but please note that this does not guarantee formal re-evaluation let alone eventual acceptance.

Reviewer #1:

This study looks at whether there are sex differences in the variability of traits in mice, via a meta-analysis of published datasets. The analyses show that females typically show greater variability in traits categorised as immunological, while males show greater variability in morphological traits. Traits related to the eye were also more variable in females. These findings are interpreted in light of evolutionary theory about greater between-individual variability in males, and greater within-individual variability in female mammals due to estrus. A online tool is provided to allow researchers to consider possible sex-specific variability in traits at the experimental design phase.

I enjoyed the paper and thought the question and conclusions were interesting. The figures are great. I am not an expert in meta-analyses, nor in mice, so my comments mostly relate to the hypotheses and discussion of the results.

1) The paper jumps about quite a bit between talking about sex differences relevant to mammals only and those that might apply to animals more generally. For example, the Introduction begins with reference to biomedical research (mammals) and the estrus hypothesis (mammals) but then introduces the "male variability" hypothesis by stating the "males are often the heterogametic sex". Given that the subject of your study is the mouse, I think it would be more logical to restrict the Introduction to mammals (i.e. explain the two hypotheses with respect to mammals). You could then include a section in the Discussion on if/why we might expect the same trends in other animals (see below also).

2) I feel that the rationale behind the two hypotheses (female estrus and male variability) could be explained better in the Introduction. i.e. *why* estrus might produce higher variability in females and *why* stronger sexual selection or male heterogamety might produce greater male variability. A few extra sentences on each would probably be enough. At the same time, I think it would be worth clarifying a priori the extent to which these hypotheses are expected to apply to different traits. Some predictions are given only in the Discussion (e.g. estrus expected to mostly affect immune response and physiology).

3) The Discussion on eco-evolutionary implications would be greatly strengthened if it included at least one specific example of how sex-specific differences in trait variability might affect the evolutionary trajectory of a population. At present, one very general hypothetical is given, but I did not find it easy to follow (disease/climate change kills more of one sex than the other –> sex ratio of the population is skewed (temporarily?) –> mating system is "influenced" –> "downstream affects on population dynamics"). It is also stated that "modelling sex difference in trait variability could lead to different conclusions compared to existing models (cf 44)". The cited study there is on Eurasian sparrowhawks. I'm not familiar with this sparrowhawk study, but perhaps it is a suitable one to highlight in more detail as a clear example? What sort of different conclusions would be expected? It's great that your paper is aiming to speak to a broad range of biologists, but I think that greater clarity in this section is needed to make ecologists and evolutionary biologists really take notice.

Reviewer #2:

Summary

There are significant methodology and interpretative concerns with this article. The analysis over stretches and does not consider the potential weaknesses. It needs to refocus on the primary question of whether there is a pattern in the sex's impact on the variance for these traits. The analysis then needs to go deeper and remove other sources of variance that could be confounding their findings.

Methodology

1) The methodology is not clear.

2) Meta-analysis is used when you don't have access to the raw data – why not use mixed effect regression models?

3) The variance summary metric is calculated for an institute and strain for data collected in multiple batches, with potential baseline shifts as the data is collected across many years. This isn't a representative metric of variability for a sex as there are multiple sources of variance impacting this metric.

4) Figure 3B and code: It is very rare for a fixed effect analysis to be justifiable. Why assume that there is no variation between the different traits when testing effect of sex? Normally you would explore sources of heterogeneity by meta regression rather than just assume it is sex differences.

5) "A previous study found that the heterogametic sex was more variable in body size". If this holds, would not traits that are correlated with body weight also demonstrate the same finding?

6) "minimum of 2 different institutes" is a very low N. Why would this give meaningful analysis? What was the minimum amount of data for a strain*centre for a trait to be included?

7) Consider the recent discussions on phenotypic plasticity and the phenotypic interaction with the environment (https://www.nature.com/articles/s41583-020-0313-3). This suggests a fixed effect model is not appropriate. The results and approach need discussing in this context.

Conclusion

1) It isn't made clear that this analysis is trying to assess the role of sex across strains and institutes.

2) There is no discussion of the potential weakness of the analysis.

3) Figure 3A:

– Why is there no discussion of measures of heterogeneity within the meta-analysis at the population level?

– Should the differences in classification as male or female biased within functional group not be assessed by a fisher exact test and the p value adjusted for multiple testing before you state an area has a difference?

4) Concern by "Notably most SD trait means also show the greater difference in trait variance" – seems to be an eyeball rather than a statistical analysis.

5) I have concerns on relating these results to power:

– These estimates are from an analysis across strains, batches and institutes looking at global behaviour in the traits. This absolute variance measure would be very different to that seen in a lab within a classic parallel group design study with one strain.

– They advocate a factorial design but suggest the powering of the sexes independently. This feeds into the misconception that to study both sexes you have to double your sample size.

6) The authors report that this analysis on mean differences was in accordance with previous studies. Not really. The differences will arise from the different approaches taken and highlights how this summary metric is losing sensitivity. The authors relate many of these changes to a difference in body size. However, the earlier published analysis, adjusted for body weight.

7) Why would the "difference in variability impact on the potential of each sex to respond to changes in specific environments"?

Reviewer #3:

This is a comprehensive meta-analysis of empirical literature on sex differences in mammalian trait variability. The authors nicely articulate competing hypotheses: "estrus-mediated variability" (which predicts higher trait variability in females because they exhibit cyclic reproductive [estrous] hormone secretion that occurs over multi-day timescales) vs. "male variability hypothesis" (which predicts higher trait variability in males because they are the heterogametic sex). Several prior meta-analyses related to this have not provided support for the estrus-mediated variability hypothesis. The analysis performed here differs significantly from prior work in that the subjects were 27,147 mice from the International Mouse Phenotyping Consortium, which generated over 2x10^6^ data points. Unlike other meta-analyses, the subjects of this analysis were therefore more systematically evaluated (9 wildtype strains across 11 labs). A total of 218 continuous traits were evaluated, grouped into 9 functional trait groups. Some traits were biased towards males and others towards females. There was no consistent pattern of greater variability in either sex. The results support a straightforward conclusion that neither hypothesis adequately explains patterns of trait variability. the discussion is a restrained defense of the practice of including females (please clarify that monitoring of estrous cycles was not performed in these studies so the females are classified as as "unstaged"); consequently females can be included in research studies without a default assumption that they are any more likely to introduce more variability than including males. The authors also apply their data on widespread differences in trait specific lnCVR values to the potential for phenotypic response to selection due to rapidly changing environmental events. The Discussion is well written with the sections that are each meaningful. The web-based tool is a very helpful contribution. The discussion of statistical implications of the work (e.g., equalizing power and Type I consequences of unequal variance) is of significance to research on mammalian biology.

1) The present work adds important new information to a growing literature (see for example Smarr BL, Rowland NE Zucker I. Male and female mice show equal variability in food intake across 4-day spans that encompass estrous cycles. PLoS One. 2019 Jul 15;14(7):e0218935) indicating that incorporation of unstaged female rodents in biomedical research does not increase variability compared to that generated by males; importantly, it also specifies several circumstances in which specific traits are more variable in one sex than the other.

2) The statement “ This higher trait variability, resulting from females being at

different stages of their estrous cycle, is the main reason for why female research subjects are often excluded from biomedical research trials, especially in the neurosciences, physiology and pharmacology” is a strong overgeneralization and should be tempered and/or clarified: "However, scientists in (bio-)medical fields have not traditionally regarded sex as a biological factor of intrinsic interest (2-7)." This is an overstatement. The study of sex differences and sexual differentiation in mammals (a class of animals of most direct relevance to biomedical research) has a long history, complete with dedicated journals (e.g. Biology of Sex Differences), learned societies, etc. Such an enduring interest in sex among biologists only makes the present work more interesting and important. This critique may be addressed with a more clear definition of "(bio-)medical", here, and throughout the manuscript.

3) Colloquialisms such as "This is an important step, but we can go much further" are vague and difficult for this reader to endorse as true, as written and we recommend deletion.

4) In the Introduction, the authors delineate competing hypotheses: "estrus-mediated variability" vs. "male variability hypothesis". In their elaboration of the former hypothesis, the authors should clarify that the historical concern regarding decreased power and increased variability in females compared to males specifically regarded the inclusion of females that were not synchronized (or "staged") so as to be tested/treated on the same day/phase of the estrous cycle. Data from these so-called “randomly cycling” females were predicted to be more variable than data from males. "Staged" females were presumed to be less variable, and the interventions and costs associated with the presumed need for staging are viewed as onerous. But a growing literature, including the important new results from the present study, argues that there is no empirical support for the contention that females generally are more variable than males across many traits.

5) Materials and methods: the data analysis pipeline is clear and rigorous. It should be stated that the data used come from unstaged females.

[Editors’ note: further revisions were suggested prior to acceptance, as described below.]

Thank you for resubmitting your work entitled "Sexual dimorphism in trait variability and its eco-evolutionary and statistical implications" for further consideration by *eLife*. Your revised article has been evaluated by a Senior Editor, a Reviewing Editor, and a reviewer.

The manuscript has been improved, but three issues were raised in this second round of review that need to be addressed before final acceptance:

1) Please ensure that the proposal for sample heterogenization by Voelkl et al., 2020, is not misrepresented in your discussion. In that review, the authors do recommend heterogenization but not in an uncontrolled fashion, rather a systematic inclusion of variation with a randomised block design. If there is only one replicate per treatment per block then, yes, the variance measured in this manuscript (and hence app) will be a good representation of the variance expected. This type of design is likely to be rare and most researchers will use a RBD with replication within a block and less strains. Please carefully revise your manuscript to avoid suggesting that the recommendation is to mix different strains (as that isn't quite correct).

2) Regarding the powering of studies, you state: "If we assume that responses to an experimental treatment will be similar between the sexes for this functional trait group, we will require more females to achieve the same statistical power as for the males." The wording here implies that the power calculations for a treatment effect are calculated separately for the males and the females. This reinforces the misconception that when you study both sexes you should consider the powering as two independent randomised complete designs (and hence if the variance is equal you would double the sample size). As you are talking about designs which include both sexes (not the selection of one versus the other), there is a need to explicitly state that power in a factorial design is achieved by assessment of the treatment effect from both the males and the females. If the variance is different between the sexes, there is a need to increase the N in the more variable sex to achieve the same final sensitivity.

3) While the Discussion section on "eco-evolutionary implications" is improved, this section still lacks clarity, the second paragraph in particular. What are "population dynamic models" and why are they important? What "different conclusions" are alluded to when you state, "explicitly modelling sex difference in trait variability could lead to different conclusions compared to traditional modelling approaches"? There is also no longer any mention of climate change in this paragraph, even though it is mentioned in the Abstract as an example of an "eco-evolutionary ramification". These points need strengthening to justify the article's title and capture the attention of a wider range of biologists.

Finally, the editors of *eLife* have recently decided to adopt the STRANGE framework for animal-behaviour research (https://www.nature.com/articles/d41586-020-01751-5?sf235295265=1) and will shortly update the journal's author guidelines and transparent reporting form. Given the link between STRANGE and the topic of your article, please consider, if possible, in your final revision how your study might engage with this new framework.

---

## [Author Response]

[Editors’ note: the authors resubmitted a revised version of the paper for consideration. What follows is the authors’ response to the first round of review.]

Reviewer #1:[…]1) The paper jumps about quite a bit between talking about sex differences relevant to mammals only and those that might apply to animals more generally. For example, the Introduction begins with reference to biomedical research (mammals) and the estrus hypothesis (mammals) but then introduces the "male variability" hypothesis by stating the "males are often the heterogametic sex". Given that the subject of your study is the mouse, I think it would be more logical to restrict the Introduction to mammals (i.e. explain the two hypotheses with respect to mammals). You could then include a section in the Discussion on if/why we might expect the same trends in other animals (see below also).

This is a great suggestion. We have changed the corresponding paragraph accordingly; now this reads:

“Second, the “greater male variability hypothesis” suggests males exhibit higher trait variability because of two different mechanisms. The first mechanism is based on males being the heterogametic sex in mammals...”

2) I feel that the rationale behind the two hypotheses (female estrus and male variability) could be explained better in the Introduction. i.e. why estrus might produce higher variability in females and why stronger sexual selection or male heterogamety might produce greater male variability. A few extra sentences on each would probably be enough. At the same time, I think it would be worth clarifying a priori the extent to which these hypotheses are expected to apply to different traits. Some predictions are given only in the Discussion (e.g. estrus expected to mostly affect immune response and physiology).

We realise that we were too concise in the original manuscript. We have added additional explanations to the estrous cycle hypothesis:

“…. A wide range of labile traits are presumed to co-vary with physiological changes that are induced by reproductive hormones. High variability is, therefore, expected to be particularly prominent when the stage of the estrous cycle is unknown and unaccounted for. This higher trait variability, resulting from females being at different stages of their estrous cycle, is the main reason for why female research subjects are often excluded from biomedical research trials, especially in the neurosciences, physiology and pharmacology. …”

And the male variability hypothesis:

“Second, the “greater male variability hypothesis” suggests males exhibit higher trait variability because of two different mechanisms. […] So far, the “greater male variability hypothesis” has gained some support in the evolutionary and psychological literature.”

Because we had no a priori expectations regarding which traits would be differently affected, this is not expanded upon in the Introduction. In fact, we had expected to find overarching support for either higher male or higher female variability, which was not supported by the data.

3) The Discussion on eco-evolutionary implications would be greatly strengthened if it included at least one specific example of how sex-specific differences in trait variability might affect the evolutionary trajectory of a population. At present, one very general hypothetical is given, but I did not find it easy to follow (disease/climate change kills more of one sex than the other –> sex ratio of the population is skewed (temporarily?) –> mating system is "influenced" –> "downstream effects on population dynamics"). It is also stated that "modelling sex difference in trait variability could lead to different conclusions compared to existing models". The cited study there is on Eurasian sparrowhawks. I'm not familiar with this sparrowhawk study, but perhaps it is a suitable one to highlight in more detail as a clear example? What sort of different conclusions would be expected? It's great that your paper is aiming to speak to a broad range of biologists, but I think that greater clarity in this section is needed to make ecologists and evolutionary biologists really take notice.

We have rewritten the entire paragraph to strengthen our eco-evolutionary implications:

“Demographic parameters, such as age-dependent mortality rate, are often different for each sex. Indeed, recognition of this fact has resulted in population dynamic models taking these widely observed sex differences into account. For example, a study on European sparrowhawks found that variability in mortality was higher in females compared to males. In this species, sex-specific variation affects age-dependent mortality and results in higher female life expectancy. As such, explicitly modelling sex difference in trait variability could lead to different conclusions compared to traditional modelling approaches.”

Reviewer #2:SummaryThere are significant methodology and interpretative concerns with this article. The analysis over stretches and does not consider the potential weaknesses. It needs to refocus on the primary question of whether there is a pattern in the sex's impact on the variance for these traits. The analysis then needs to go deeper and remove other sources of variance that could be confounding their findings.

Reviewer 2 has carefully read the code and its annotations, because s/he correctly points out our mistake in the annotation (see below; this html is an extensive document including a flow diagram of our data processing and statistical analyses based on >2,900 lines of code). Thus, we believe that reviewer 2’s concerns mainly stem from our omissions of some methodological descriptions and justifications in the original manuscript.

We have taken these criticisms seriously to further improve the clarity of our written methodology. In our revised version, we have clearly stated how our method deals with the confounders (see our replies below). By using a meta-analytical method (2nd comment by reviewer 2 – and we have expanded it there), many confounding effects have been taken care of. We do note, however, that our extended html was indeed helpful because reviewer 3 states “Materials and methods: the data analysis pipeline is clear and rigorous.”, which contrasts with what reviewer 2 says.

Methodology1) The methodology is not clear.

We have endeavoured to make our methodology more accessible and comprehensive by addressing reviewer 2’s and the others’ comments. We would like to point out that we do have a methodological flowchart in our supplementary material as we were very much aware that our method is complex (and therefore, it might have been unclear). We have added more explanation in the main text and the supplement where possible (see our other replies to reviewer 2 below).

2) Meta-analysis is used when you don't have access to the raw data – why not use mixed effect regression models?

This is an important comment; it appears that our original version did not cover the justification for the approach clearly enough. However, in medicine, for example, a meta-analytic method like ours are often used even when one has raw data from different studies (i.e. individual patient data (IPD) meta-analyses, where they could also analyse raw data in these “meta-analyses”, rather than using summary effect sizes; see Debray et al., 2015, 6: p293, Res. Syn. Meth.). That aside, we have three main reasons why we used a meta-analytic approach (although the original manuscript omitted the second and third reasons – we have added this now, for exact wording see below).

First, as we wrote in the original manuscript: “Our meta-analytic approach allows easy interpretation and comparison with earlier and future studies.” This is because lnCVR and lnRR can be interpreted in terms of % differences between sexes (most of us have good intuitions for % differences compared to absolute differences in particular units, say, cm).

Second, and this is probably the most important reason, in the most common versions of mixed effect models, we cannot compare and contrast differences in variances between two groups. One possible way around this would be to model heteroskedasticity explicitly (see Cleasby and Nakagawa, 2011), which would require the specification and estimation of complex variance-covariance structure in the model residuals and introduce potential model identification problems. In any case, the approach we took using lnCVR, allows for a simpler and direct test of the research question. This is the main reason to use a meta-analytic approach.

Third, the use of standardised effect size, such as lnRR (log response ratio) or lnCVR, is sometimes referred to as the “contrast” method. This is because these effect sizes are the ratio of two effects between the treatment and control groups (females and males in our case). Taking a ratio has major advantages over modelling raw data, because it controls for different units across traits and it better controls for temporal and spatial changes in the traits themselves across control/treatment groups (here males and females; assuming that changes are relative). This is one of the biggest benefits of meta-analysis.

This “contrast” approach also addresses another part of the comment made by reviewer 2 in their opening paragraph that “It needs to refocus on the primary question of whether there is a pattern in the sex's impact on the variance for these traits. The analysis then needs to go deeper and remove other sources of variance that could be confounding their findings”. By taking the ratio within each study (i.e. relative to a concurrent control under matching conditions), the confounding effects are accounted for within each study and the effect size – and any potential effect on variance – is isolated. Thus, many experiment-specific factors do not need to be modelled explicitly; this is one of the arguments underlying conventional meta-analysis. The method used here (lnCVR) uses the same “contrast” approach and has the same advantages.

We now understand that our omissions of the second and third reasons made it difficult for reviewer 2 (and others) to assess the validity of our approach, generating concerns. Therefore, we have added these reasons to the manuscript:

“… Further, the proposed method using lnCVR (and lnVR) is probably the only practical method to compare variability between two sexes within and across studies, as far as we are aware. Also, the use of a ratio (i.e. lnRR, lnVR, lnCVR) between two groups (males and females) naturally controls for different units (e.g., cm, g, ml) and also for changes in traits over time and space.”

3) The variance summary metric is calculated for an institute and strain for data collected in multiple batches, with potential baseline shifts as the data is collected across many years. This isn't a representative metric of variability for a sex as there are multiple sources of variance impacting this metric.

As described above, lnCVR (like lnRR, lnVR, and Hedges’ g) is a contrast metric that is relative to a concurrent control under matching conditions. Assuming “potential baseline shifts” affect males and females equally, these variabilities are already taken care of when using ratio-based effect sizes. We now have mentioned this point in the Introduction (see above). Also, we had added this sentence in the Materials and methods section:

“As mentioned above, the use of ratio-based effect sizes such as lnCVR, lnVR and lnRR controls for baseline changes over time and space, assuming that these changes affect males and females similarly. However, we acknowledge that we could not test this assumption.”

4) Figure 3B and code: It is very rare for a fixed effect analysis to be justifiable. Why assume that there is no variation between the different traits when testing effect of sex? Normally you would explore sources of heterogeneity by meta regression rather than just assume it is sex differences.

We agree with reviewer 2 and we have to apologise for our mistake. We believe reviewer 2 thought we might have used fixed-effect models from the code annotation:

“# Final fixed effects meta-analyses within grouping terms”. This was an annotation mistake.

In the code, we ran random-effects meta-analyses throughout. Indeed, in our Materials and methods section of the original manuscript, we stated that we have used “random-effects” models and also multilevel models, which are a version of random-effects models (see Nakagawa et al., 2017).

“…, we estimated overall effect sizes for nine functional groups by aggregating meta-analytic results via a “classical” random-effect models using the function rma.uni in the R package *metafor*.”

This is for the second-order meta-analysis aggregating traits into groups of traits:

“we used the function *rma.mv* in the R package *metafor* (Viechtbauer, 2010) by fitting the following multilevel meta-analytic model, an extension of random-effects models (sensu Nakagawa & Santos 2012):

ES_i_ ~ 1 + (1 |Strain_j_ ) + (1 | Location_k_) + (1 | Unit_i_) + Error_i_, ”

This is for each trait. As you can see, we are marginalising over strains and locations to get the average effect sizes for both lnRR (mean difference) and lnCVR (relative variance difference).

We thank reviewer 2 for spotting this mistake in code annotation, and now it has been corrected.

5) "A previous study found that the heterogametic sex was more variable in body size". If this holds, would not traits that are correlated with body weight also demonstrate the same finding?

Reviewer 2 would be correct if we were talking about mean differences (i.e. lnRR). We would expect, for example, traits, which correlate with body size, would show males have larger trait value in these traits. And this can be shown in our results of lnRR.

However, this will not necessarily be the case for differences in relative variance (i.e. lnCVR). This is because trait CVs do not necessarily correlate with trait means. This is the reason why we have provided % differences of CV (along with mean and SD) for all the traits we investigated via a Shiny App. The Shiny App will be useful for researchers to find which traits have potentially sex-biased CVs.

6) "minimum of 2 different institutes" is a very low N. Why would this give meaningful analysis? What was the minimum amount of data for a strain*centre for a trait to be included?

We are sorry that this was not clear. Despite having a minimum of two institutions, these institutions usually had replicated samples of mice from different experiments. In fact, we meta-analysed traits with between 2–18 effect sizes (mean = 9.09 effects, SD = 4.47); note that for Cochrane reviews, the median number of effect sizes per meta-analysis is 3 (so overall our meta-analyses have higher sample sizes than Cochrane reviews). While a minimum of N = 6 mice were used to create effect sizes for any given group (male or female), in reality samples sizes of male / female groups were much larger (males: mean = 396.66 (SD = 238.23), median = 465.56; females: mean = 407.35 (SD = 240.31), median = 543.89). We have now clarified these details in the Materials and methods as follows:

“Overall, we meta-analysed traits with between 2–18 effect sizes (mean = 9.09 effects, SD = 4.47). However, each meta-analysis contained a total number of individual mice that ranged from 83/91 to 13467/13449 (males/females). While a minimum of N = 6 mice were used to create effect sizes for any given group (male or female), in reality samples sizes of male / female groups were much larger (males: mean = 396.66 (SD = 238.23), median = 465.56; females: mean = 407.35 (SD = 240.31), median = 543.89).”

In addition, we want to emphasize that using meta-analytic methods accounts for sampling error variances in estimating overall mean effect sizes.

7) Consider the recent discussions on phenotypic plasticity and the phenotypic interaction with the environment (https://www.nature.com/articles/s41583-020-0313-3). This suggests a fixed effect model is not appropriate. The results and approach need discussing in this context.

As mentioned above, we did not use fixed-effect models; this was an annotation mistake left in our code, which is now fixed (see our reply to Comment 4). Further, we are aware of this paper. Indeed, several co-authors of the paper are long-standing collaborators of the senior author. We have now cited this paper in the manuscript (the reason for this is described below).

Conclusion1) It isn't made clear that this analysis is trying to assess the role of sex across strains and institutes.

Reviewer 2 is correct. Our models marginalised the effect of strains and institutions when estimating average effect size (like many meta-analyses would do). However, measures of total heterogeneity (i.e. the sum of total strain, production center and unit level variance / total variance) for each trait were extremely low (0-1% lnCVR; 0-0.4% lnVR and 0-0.4% lnRR). We now make this clear by stating the following in the Discussion:

“It is important to know that for each trait we obtained the mean effect size (i.e. lnCVR) over strains and locations. As such, our results may not necessarily apply to every group of mice, which may or may not result in stronger support for either of the two hypotheses.”

Please also see our other replies related to the limitations of our work below.

2) There is no discussion of the potential weakness of the analysis.

Now we have added sentences discussing potential limitations (see our replies to other comments; points 2 and 3). We also, however, discussed the strength of our analyses too in these locations. Our work will, of course, have all the typical weaknesses of meta-analyses, although we do not expect publication biases, which is excellent news.

Indeed, our work was motivated by the paper by reviewer 3 (cited in the original manuscript):

B. J. Prendergast, K. G. Onishi, I. Zucker, Female mice liberated for inclusion in

neuroscience and biomedical research. Neurosci Biobehav Rev 40, 1–5 (2014).

This important synthesis comparing CV between females and males was not a formal meta-analysis. Therefore, this synthesis does not account for the differences in sample sizes (the number of mice) and baseline changes in traits. Our meta-analytic approach goes beyond this paper by statistically formalising how one can compare CV between two groups (males vs. females). This must be the reason, we believe, reviewer 3 is very favourable and liked our approach, as suggested by his comments. Our responses to reviewer 2’s comments 2 and 3 have now dealt with this point, highlighting the strengths as well as weaknesses of our approach.

3) Figure 3A:– Why is there no discussion of measures of heterogeneity within the meta-analysis at the population level?

This is because this was not particularly relevant to our main aim – testing the two hypotheses explaining sex differences in variability. In addition, our analysis focused on broad trait categories, because at the trait level, effect sizes ranged between 218 /trait making estimating heterogeneity challenging to compute. We have, however, calculated these as requested by reviewer 2 and, unsurprisingly, measures of total heterogeneity for each trait (218 traits total) were extremely low (ranges 0 – 1% lnCVR; 0 – 0.4% lnVR and 0 – 0.4% lnRR). Given heterogeneity for individual traits is unlikely to be reliable, and is not of direct interest, we have not included them in our revision. However, should the Editor or reviewer feel these are important we are still happy to provide them.

– Should the differences in classification as male or female biased within functional group not be assessed by a fisher exact test and the p value adjusted for multiple testing before you state an area has a difference?

This suggestion probably stemmed from our lack of explanation for the reason why we did not do such tests. The main reason is that, if you apply such statistics as Fisher’s exact tests, we are endorsing vote counting practices. Vote counting (statistical tests of count data) has been severely criticised because it does not take sample sizes (the number of subjects) into account (see, for example, Higgins and Green, 2018; Cochrane Handbook). Instead, we did provide statistical inferential tests for corresponding meta-analyses, which are recommended. Further, we did not do such statistical tests because Fisher’s exact tests or related tests (Chi-square tests) are severely limited by sample sizes. Everything else being equal, higher sample sizes will eventually bring statistical significance, as we described in Nakagawa and Cuthill (2007, 82:p591, Biol Rev).

We have added the following in the Materials and methods section:

“Although we present the frequencies of male- and female-biased traits in Figure 3A, we did not run inferential statistical tests on these counts because such tests would be considered as vote-counting, which has been severely criticised in the meta-analytic literature…”

Nonetheless, if the Editor and reviewers think providing inferential statistics for these counts (i.e. vote counting) would be helpful additions, we would be happy to add them.

4) Concern by "Notably most SD trait means also show the greater difference in trait variance" – seems to be an eyeball rather than a statistical analysis.

Yes, this was just a description of the counts because of the reason described above in relation to statistical tests of counts without considering underlying sample sizes (i.e. vote counting) – these inferences are however statistically supported by meta-analytic results, which we have now made clear in the manuscript.

5) I have concerns on relating these results to power:– These estimates are from an analysis across strains, batches and institutes looking at global behaviour in the traits. This absolute variance measure would be very different to that seen in a lab within a classic parallel group design study with one strain.

We understand reviewer 2’s concern here. Our results are overall effects and it may not be readily applicable to a specific strain. However, we would like to point out three reasons why we have done this. First, our main aim is to compare sex differences in trait variability in order to test two competing hypotheses in a general manner that is not specific to traits or labs (of course, we apologise that we omitted these reasons in the original manuscript, which has led to this misunderstanding).

Second, reviewer 2 coincidentally points out this paper in the comment above:

Voelkl B, Altman NS, Forsman A, Forstmeier W, Gurevitch J, Jaric I, Karp NA, Kas MJ, Schielzeth H, Van de Casteele T, Würbel H.

Reproducibility of animal research in light of biological variation. Nature Reviews Neuroscience. 2020 Jun 2:1-0.

According to this paper, they are encouraging future experiments to use techniques called “heterogenization” where different strains are mixed to increase the robustness of experimental results. In this very context, our estimates of differences in mean traits and SD and CV are, we believe, entirely relevant.

Third, as reviewer 2 correctly indicates, we also believe that, if researchers were to use one strain of mice, it would be more useful to use strain- or lab-specific estimates of (descriptive) statistics (mean, SD and CV) from that particular lab and strain; note that for common traits such data are usually made available by commercial breeding facilities. However, our results are overall means, and so, we are providing the very first benchmarks for researchers to compare their statistics to. Given these reasons, our results are widely relevant as we described in the original manuscript.

However, it is important to address this concern by reviewer 2, so that we have now added:

“Further, these estimates are overall mean differences across strains and locations. Therefore, these may not be particularly informative if one’s experiment only includes one specific strain. However, we point out that our estimates may be useful in the light of a recent recommendation of using “heterogenization” where different strains are mixed to increase the robustness of experimental results (Voelkl et al., 2020). Also, even in the case of using a particular strain, our tool can provide potentially useful benchmarks.”

– They advocate a factorial design but suggest the powering of the sexes independently. This feeds into the misconception that to study both sexes you have to double your sample size.

We certainly did not intend to suggest researchers need to double their sample sizes. Indeed, we did not mention anything about “doubling sample size in both sexes” in our previous manuscript version. Rather, we indicated that researchers may want to consider how they allocate sampling effort for each of the sex to maximise power, as follows:

“For example, given a limited number of animal subjects in an experiment measuring immunological traits, a balanced sex ratio may not be optimal. Female immunological traits are generally more variable (i.e. higher CV and SD). If we assume that responses to an experimental treatment will be similar between the sexes for this functional trait group, we will require more females to achieve the same statistical power as for the males.”

However, reviewer 2 is correct in the context that adequately powered experiments require a lot of animals. And this will be made even more clear, we believe, if researchers start conducting power analysis separately for both sexes.

6) The authors report that this analysis on mean differences was in accordance with previous studies. Not really. The differences will arise from the different approaches taken and highlights how this summary metric is losing sensitivity. The authors relate many of these changes to a difference in body size. However, the earlier published analysis, adjusted for body weight.

We have sought to make our discussion on these points more accurate. The paper which looked at mean differences between sexes is:

Karp, N. A., et al. "Prevalence of sexual dimorphism in mammalian phenotypic traits." Nature communications 8.1 (2017): 1-12.

In this paper, the authors both looked at mean differences *with* and *without* controlling for weights. Therefore, we made it more accurate by saying:

“In general, we found many traits to be sexually dimorphic (Figure 4) in accordance with the previous study, which used the same database (Karp et al., 2017), although the original study did provide estimates for sex differences in traits both with and without controlling for weight (we did not control for weight; cf. 40).”

This question also relates to why we do not use weight correction in our analyses. There are two main reasons for this.

First, our focus on the paper is to compare trait variability differences between sexes, not residual variability differences between sexes once weight is controlled for. Such residual analyses (or related analyses) have several potential shortcomings (e.g., Garcia-Berthou 2001, p708, J Anim Ecol). Also, some of us have written about the dangers of controlling weights when comparing two groups in this paper:

Nakagawa, S., et al. "Divide and conquer? Size adjustment with allometry and intermediate outcomes." BMC biology 15.1 (2017): 1-6.

Second, we were and are interested in actual variability differences across different traits, which are more suitable for testing, we believe, the two hypotheses in our manuscript. Also, such comparisons were also made in reviewer 3’s important original synthesis comparing CV between males and females (listed above).

7) Why would the "difference in variability impact on the potential of each sex to respond to changes in specific environments"?

This may have not been very clear in the original version. What we wanted to say is that, all else being equal (e.g., the same trait means), the sex which has higher trait variability is more likely to be under stronger selection than the other sex. This point was not well articulated. We have fully rewritten this paragraph:

“Demographic parameters, such as age-dependent mortality rate (Lemaitre et al., 2020), are often different for each sex. Indeed, recognition of this fact has resulted in population dynamic models taking these widely observed sex differences into account (Colchero et al., 2017; Caswell and Weeks, 1986). For example, a study on European sparrowhawks found that variability in mortality was higher in females compared to males (Lindstrom and Kokko, 1998). In this species, sex-specific variation affects age-dependent mortality and results in higher female life expectancy. As such, explicitly modelling sex difference in trait variability could lead to different conclusions compared to traditional modelling approaches.”

Reviewer #3:[…]1) The present work adds important new information to a growing literature (see for example Smarr BL, Rowland NE Zucker I. Male and female mice show equal variability in food intake across 4-day spans that encompass estrous cycles. PLoS One. 2019 Jul 15;14(7):e0218935) indicating that incorporation of unstaged female rodents in biomedical research does not increase variability compared to that generated by males; importantly, it also specifies several circumstances in which specific traits are more variable in one sex than the other.

We thank the reviewer for his positive feedback. We have now further extended our explanations on the estrous cycle, such as in:

“… A wide range of labile traits are presumed to co-vary with physiological changes that are induced by reproductive hormones. High variability is, therefore, expected to be particularly prominent when the stage of the estrous cycle is unknown and unaccounted for. This higher trait variability, resulting from females being at different stages of their estrous cycle, …”

In addition, we have added a recent publication by the reviewer to our manuscript, which predicts the adverse effects of sex-specific drug testing:

Zucker, I., Prendergast, B.J. Sex differences in pharmacokinetics predict adverse drug reactions in women. Biol Sex Differ 11, 32 (2020).

The respective part now reads:

“For example, we know far more about drug efficacy in male compared to female subjects, contributing to a poor understanding of how the sexes respond differently to medical interventions (Nowogrodzki, 2017). This gap in knowledge is predicted to lead to overmedication and adverse drug reactions in women (Zucker and Prendergast, 2020).”

2) The statement “This higher trait variability, resulting from females being atdifferent stages of their estrous cycle, is the main reason for why female research subjects are often excluded from biomedical research trials, especially in the neurosciences, physiology and pharmacology” is a strong overgeneralization and should be tempered and/or clarified: "However, scientists in (bio-)medical fields have not traditionally regarded sex as a biological factor of intrinsic interest." This is an overstatement. The study of sex differences and sexual differentiation in mammals (a class of animals of most direct relevance to biomedical research) has a long history, complete with dedicated journals (e.g. Biology of Sex Differences), learned societies, etc. Such an enduring interest in sex among biologists only makes the present work more interesting and important. This critique may be addressed with a more clear definition of "(bio-)medical", here, and throughout the manuscript.

We are well aware that in the biological field and biomedically relevant fields sex differences are a central research topic, and were trying to imply with our wording of “(bio-)medical” (i.e. main concern on medical applications, less interest in the biomedical study species, excluding “biology” as a field), this has only gained recent attention (i.e. within the last 10 years there has been a marked increase in journals specialising in this topic). To tone this down we have changed that sentence, as follows:

“… However, scientists in many (bio-)medical fields have not necessarily regarded sex as a biological factor of intrinsic interest …”

3) Colloquialisms such as "This is an important step, but we can go much further" are vague and difficult for this reader to endorse as true, as written and we recommend deletion.

This sentence has been deleted.

4) In the Introduction, the authors delineate competing hypotheses: "estrus-mediated variability" vs. "male variability hypothesis". In their elaboration of the former hypothesis, the authors should clarify that the historical concern regarding decreased power and increased variability in females compared to males specifically regarded the inclusion of females that were not synchronized (or "staged") so as to be tested/treated on the same day/phase of the estrous cycle. Data from these so-called “randomly cycling” females were predicted to be more variable than data from males. "Staged" females were presumed to be less variable, and the interventions and costs associated with the presumed need for staging are viewed as onerous. But a growing literature, including the important new results from the present study, argues that there is no empirical support for the contention that females generally are more variable than males across many traits.

We thank the reviewer for his detailed account of staging and its predictions for female variability. Indeed, this is exactly what we intended to convey. We have clarified our writing, and the new paragraph now reads:

“First, the “estrus-mediated variability hypothesis” (Figure 2), which emerged in the (bio-)medical research field, assumes that the female estrous cycle (see for example 6, 18) causes higher variability across traits in female subjects. A wide range of labile traits are presumed to co-vary with physiological changes that are induced by reproductive hormones. High variability is, therefore, expected to be particularly prominent when the stage of the estrous cycle is unknown and unaccounted for. This higher trait variability, resulting from females being at different stages of their estrous cycle, is the main reason for why female research subjects are often excluded from biomedical research trials, especially in the neurosciences, physiology and pharmacology…”

5) Materials and methods: the data analysis pipeline is clear and rigorous. It should be stated that the data used come from unstaged females.

We added: “… All data are from unstaged females (with no information about the stage of their estrous cycle)...”

[Editors’ note: what follows is the authors’ response to the second round of review.]

The manuscript has been improved, but three issues were raised in this second round of review that need to be addressed before final acceptance:1) Please ensure that the proposal for sample heterogenization by Voelkl et al., 2020, is not misrepresented in your discussion. In that review, the authors do recommend heterogenization but not in an uncontrolled fashion, rather a systematic inclusion of variation with a randomised block design. If there is only one replicate per treatment per block then, yes, the variance measured in this manuscript (and hence app) will be a good representation of the variance expected. This type of design is likely to be rare and most researchers will use a RBD with replication within a block and less strains. Please carefully revise your manuscript to avoid suggesting that the recommendation is to mix different strains (as that isn't quite correct).

This is an important point. We changed this to:

“However, we point out that our estimates may be useful in the light of a recent recommendation of using “heterogenization” where many different strains are systematically included (i.e., randomized complete block design) to increase the robustness of experimental results (Voelkl et al., 2020). However, note that an experiment with heterogenization might only include a few strains with several animals per strain. Even in such a case using just a few strains, our tool could provide potentially useful benchmarks.”

2) Regarding the powering of studies, you state: "If we assume that responses to an experimental treatment will be similar between the sexes for this functional trait group, we will require more females to achieve the same statistical power as for the males." The wording here implies that the power calculations for a treatment effect are calculated separately for the males and the females. This reinforces the misconception that when you study both sexes you should consider the powering as two independent randomised complete designs (and hence if the variance is equal you would double the sample size). As you are talking about designs which include both sexes (not the selection of one versus the other), there is a need to explicitly state that power in a factorial design is achieved by assessment of the treatment effect from both the males and the females. If the variance is different between the sexes, there is a need to increase the N in the more variable sex to achieve the same final sensitivity.

Now we understood this point. We have changed the original paragraph:

“For example, given a limited number of animal subjects in an experiment measuring immunological traits, a balanced sex ratio may not be optimal. Female immunological traits are generally more variable (i.e. higher CV and SD). If we assume that responses to an experimental treatment will be similar between the sexes for this functional trait group, we will require more females to achieve the same statistical power as for the males.”

To this revised paragraph:

“For example, female immunological traits are generally more variable (i.e. having higher CV and SD). Therefore, in an experiment measuring immunological traits, we would need to include a larger sample (*N*) of females than males (*N*_[female]_ > *N*_[male]_; *N*_[total]_ = *N*_[female]_ + *N*_[male]_) to achieve the same power as when the experiment only includes males (*N*_[total*]_ = 2*N*_[male]_). In other words, this experiment with both sexes would need a larger sample size than the same experiment with males only (*N*_[total]_ > *N*_[total*]_).”

3) While the Discussion section on "eco-evolutionary implications" is improved, this section still lacks clarity, the second paragraph in particular. What are "population dynamic models" and why are they important? What "different conclusions" are alluded to when you state, "explicitly modelling sex difference in trait variability could lead to different conclusions compared to traditional modelling approaches"? There is also no longer any mention of climate change in this paragraph, even though it is mentioned in the Abstract as an example of an "eco-evolutionary ramification". These points need strengthening to justify the article's title and capture the attention of a wider range of biologists.

To address all these points, we have reworded this original paragraph to this:

“Demographic parameters, such as age-dependent mortality rate (Lemaître et al., 2020) can often be different for each sex. For example, a study on European sparrowhawks found that variability in mortality was higher in females compared to males (Colchero et al., 2017). In this species, sex-specific variation affects age-dependent mortality and results in higher average female life expectancy. Therefore, population dynamic models, which make predictions about how populations change in their size over time, should take sex differences in variability into account to produce more accurate predictions (cf. Caswell and Wekks, 1986; Lindstrom and Kokko, 1998). In our rapidly changing world, better predictions on population dynamics are vital for understanding whether climate change is likely to result in population extinction and lead to further biodiversity loss.”

Also, in the preceding paragraph, we added this sentence to expand our eco-evolutionary ramifications and the link to climate change:

“For example, more variable morphological traits of males could potentially provide them with better capacity than females to adapt morphologically to changing climate.”

Finally, the editors of eLife have recently decided to adopt the STRANGE framework for animal-behaviour research (https://www.nature.com/articles/d41586-020-01751-5?sf235295265=1) and will shortly update the journal's author guidelines and transparent reporting form. Given the link between STRANGE and the topic of your article, please consider, if possible, in your final revision how your study might engage with this new framework.

Our original manuscript has a relevant sentence:

“Therefore, this sex difference in variability could be more pronounced under natural conditions compared to laboratory settings. This relationship may explain why male-biased morphological traits are larger and more variable.”

Further, we have added this sentence citing the STRANGE framework paper:

“Incidentally, heterogenization would be key to make one’s experimental outcome more generalizable (Webster and Rutz, 2020). ”

However, we note that we do not mention the term STRANGE in our paper, as introducing this framework requires a good explanation, which we feel is out of the scope of this manuscript.